# Using Machine Learning to Explore the Crucial Factors of Assistive Technology Assessments: Cases of Wheelchairs

**DOI:** 10.3390/healthcare10112238

**Published:** 2022-11-09

**Authors:** Kwo-Ting Fang, Ching-Hsiang Ping

**Affiliations:** Department of Information Management, National Yunlin University of Science and Technology, Yunlin 640, Taiwan

**Keywords:** decision tree, machine learning, assistive technology, wheelchair

## Abstract

The global population is gradually entering an aging society; chronic diseases and functional disabilities have increased, thereby increasing the number of people with limitations. Therefore, the demand for assistive devices has increased substantially. Due to numerous and complex types of assistive devices, an assessment by a professional therapist is required to help the individual find a suitable assistive device. According to actual site data, the assessment needs of “wheelchairs” accounted for most of the cases. Therefore, this study identified five key evaluation characteristics (head condition, age, pelvic condition, cognitive ability, and judgment) for “transit wheelchairs” and “reclining and tilting wheelchairs” from the diagnostic records of “wheelchairs” using the classification and regression trees (CART) decision tree algorithm. Furthermore, the study established an evaluation model through the Naïve Bayes classification method and obtained an accuracy rate of 72.0% after a 10-fold cross-validation. Finally, the study considered users’ convenience and combined it with a LINE BOT to allow the user or the user’s family to engage in self-evaluation. Preliminary suggestions for wheelchair types were given through the assessment model so that evaluators could not only determine a case’s situation in advance and reduce the time required for fixed-point or home assessments, but also help cases find the appropriate wheelchair type more easily and quickly.

## 1. Introduction

The pace of an increasing aging population is an issue of great concern worldwide, particularly in Taiwan. According to the World Population Prospects (WPP) 2019 annual report, the proportion of the global population over the age of 65 will be close to 12.0% in 2030, and it is estimated to reach 16.0% in 2050 [1]. In particular, Taiwan joined the “aging society” in 2018 and is expected to enter a “super-aged society” in 2026. This means that the older people population over the age of 65 will account for 20.0% of the total population [2]. This aging trend has also increased the rate of chronic diseases and physical and mental disabilities. According to the Long-term Care Plan 2.0 policy in Taiwan, the disability rate of people over the age of 65 is 12.7%. It is estimated that there will be 600,000 people with disabilities over the age of 65 in 2026 [2]. With the emergence of an aging society, Taiwan is continuously facing the dual dilemma of a substantial increase in the demand for assistive devices and a shortage of caring manpower. Assistive devices can reduce the caregiver’s burden, enhance the autonomous mobility of care recipients and delay disability [3,4]. Given recent trends and forecasting, it is clear that providing professional, comprehensive, and convenient assistive device services for care recipients and caregivers is imperative.

The Social Welfare Department reported in 2019 that the assessment of assistive devices in the assistive device center increased by 38.9% and applying for assistance from the Department for assistive device subsidies increased by 30.3% during the period of 2018. Among different devices, the demand for the assessment of “wheelchairs” accounted for the majority of cases [5]. Concomitant with the marked increase in the demand for assistive devices, there is currently an insufficient amount of manpower and resources for assistive device evaluation. Furthermore, new assistive device evaluators lack actual practical experience and most assistive device evaluators traditionally use ‘rule of thumb’ and ‘trial and error’ methods of reasoning in their evaluation. Such assessment methods may lead to ‘unforeseeable’ discomfort to individuals using an assistive device [6]. In addition to the shortage of assistive device evaluators and the lack of practical experience for new recruits, the case’s situation can only be understood through telephone consultation in the outreach assessment (e.g., home-based and fixed-point evaluations). However, there is currently no set of standard questions and answers for assessments, so it is difficult to describe the status of the case clearly. Thus, preparing an appropriate type of assistive device for a case to try out is now a critical task for evaluators, but it also leads to the possibility of re-assessment and increases the assessment time and service labor costs.

Given the in-depth interview with the director of an assistive device center on the current problems faced by the institution, Director Lin pointed out the following:

“*There are three assistive device assessors in our center, and their usual business volume is enormous. In addition to the main evaluation business, they also provide consultation, assistance in applying for subsidies, inspection, training, and follow-ups. After the launch of Long-term Care 2.0 two years ago, they must also perform fixed-point and on-site evaluation services. The manpower is obviously insufficient, and it is almost impossible to sustain all operations.*”

Based on the above problems, the main objectives of this study were expected to proceed along two paths: the first was to determine the decision-making context of different wheelchair types through the decision tree algorithm. Each assessment item in the wheelchair assessment report (Appendix A) was regarded as a deciding factor, and the wheelchair type was regarded as the classification result. The study hoped to learn about the combination of assessment items corresponding to each wheelchair type and provide a rough direction for newcomers with assistive device assessment for reference.

The second important path of the study was to build a “wheelchair” assistive device evaluation model through machine learning algorithms to predict the appropriate wheelchair type for a case. Taking a predictability standpoint, the assistive device evaluation model was combined with a LINE BOT, an embedded AI chatbot, so that “cases or their family members” could report their personal situation to the chatbot system via the questions provided by the system. Based on their reported personal situation, noteworthy suggestions on the types of assistive devices through the evaluation model could be given and sent to the evaluator. By referring to the type suggestions given by the chatbot system and selecting the appropriate type for the case to try out, evaluators can not only reduce the time for field assessment preparation, but also help cases find the appropriate wheelchair type more easily and quickly.

## 2. Literature Discussion

### 2.1. Assistive Technology

In the context of a rapidly aging social structure, the population in need of long-term care is increasing yearly and it will be an issue that will continuously dominate society in the future. In order to maintain essential life functions, older people with long-term care needs often tend to rely on the care and services of others. The emergence of assistive technology has enabled the people with disabilities to improve their lives and even complete more complex tasks independently [4,7].

Assistive Technology (AT) refers to services, equipment, customized products, or related businesses established to increase or maintain the operational functions of the people with disabilities [8]. In general, AT includes AT devices and AT services. AT devices are products or equipment used to help the people with disabilities maintain or improve their daily activities, such as wheelchairs, prosthetics, and urinals. The definition of AT Services encompasses all services that support the people with disabilities in the selection, acquisition, and use of AT equipment [9]. Therefore, AT requires a complete product system and a proper assisting service to help the people with disabilities find suitable products [10].

### 2.2. Assistive Technology Services

Basically, assistive device services not only include consultation, assessment, acquisition, usage training, tracking, maintenance, and adjustment, but also include assistive device recycling, rental, display, and promotion. Social and Family Department [3] indicated that the growth rate of assistive devices assessment was as high as 38.0%, the highest growth rate among all assistive devices. This shows that the demand for assistive devices assessment is increasing, and it echoes a shift toward extending services, such as fixed-point, or home visits, in Taiwan’s aging society.

According to the Social Welfare and Family Department’s report on assistive device services, “personal mobility assistive devices” accounted for 32.7% of the number of assistive device subsidies for the people with disabilities in municipalities and counties (cities) nationwide. One out of every three assessment subsidy cases was for personal mobility aid, followed by “furniture and modified components for homes and other places”. In particular, wheelchairs account for the largest proportion with respect to “personal mobility assistive devices” [5]. For validation purpose, this study also included interviews with the director of the assistive devices center and ascertained that there was in fact a great demand for wheelchair assessments as reported.

Machine learning has recently garnered significant attention from the medical care industry. It not only helps physicians speed up the diagnostic process and but also assists them in identifying critical features of the causes of ailments or disabilities [11]. The process followed by an assessment assistive device and medical diagnosis by a professional is similar: both conduct diagnosis based on the individual’s physical condition, medical condition, and medical history. However, previous studies have rarely mentioned the use of machine learning in terms of preparing assessment assistive devices for diagnosis.

### 2.3. Machine Learning for Healthcare Diagnosis

Based on a retrospective survey of machine learning for medical diagnosis study, in terms of five diseases (heart disease, diabetes, liver disease, dengue, and hepatitis), Fatima and Pasha [11] pointed out that Support Vector Machine (SVM), Naïve Bayes (NB), and Bayesian Network (BN) algorithms are widely used in disease diagnosis.

SVM, created by Vladimir Vapnik, is a supervised algorithm [12]. The concept is to find a decision boundary that maximizes the margins of the two categories to achieve optimal classification [13]. It is suitable for numerical data and can handle nonlinear and binary classification problems [11]. Conversely, assuming the sample features are independent, NB mainly applies conditional probability for classification prediction, which is suitable for datasets with category types and multi-class prediction problems [14]. BN is an algorithm based on Naïve Bayesian extensions and its application depends on professional knowledge in the field [15]. The difference is that the conditional probabilities are computed together for attributes that have dependencies before training, thereby forming a network, and no conditional independence is assumed for BN.

By and large, in medical care, doctors or other medical staff are often faced with situations that require diagnostic decisions, such as the prescription of drugs to patients, assistive device evaluators assessing the specifications of assistive devices required by people with disabilities, and disease diagnosis. However, in fact, these evaluation and decision-making criteria mainly stem from professional knowledge and practical experience and not so much from clinical data for diagnosis [16]. Specifically, new medical personnel who lack practical experience, often inevitably, need to spend much time searching for information when making diagnostic decisions, resulting in prolonged diagnosis time and misdiagnosis.

In order to assist and speed up doctors’ diagnoses, decision trees, in terms of context visualization, are widely used in medical diagnosis [17,18,19,20]. Using classification and regression trees (CART) to classify and diagnose hepatitis, Sathyadevi [21] presented an accuracy rate of 83.2%. She further compared the ID3 and C4.5 and found that the accuracy rate was 64.8% and 71.4%, respectively. In a similar manner, Saxena and Sharma [22] utilized C4.5 to predict and classify patients with cardiovascular disease (which were divided into five different types) and reached an accuracy rate of 86.0%. In 2019, Maji and Arora [23] posited that C4.5 could predict cardiovascular disease with an accuracy of 76.7%.

In terms of achieving better execution speed, Hashi et al. [13] compared K-Nearest Neighbor (KNN) and C4.5 in their ability to establish a regular pattern of diabetes and predict whether a patient has diabetes. KNN reached an accuracy rate of 77.0% while C4.5 reached an accuracy rate of 90.4%. Additionally, Jiang [24] adopted the K-mean to divide the data into five groups based on high to low income and then further used C5.0 to predict the type of ward reservation by the patient. Jiang’s study accomplished an accuracy rate of 66.4%.

The random forest algorithm, with the characteristics of strong classification, predictive ability, good performance, and clear feature attribute seems to offer promising results for making dramatic progress and wide strides in different fields. Moreover, compared with other machine learning algorithms, such as SVM and artificial neural network (ANN), it has better explanatory power [25]. Based on the genetic algorithm for feature selection, Azar et al. [26] reported an accuracy rate of 92.2% when they utilized the random forest algorithm to diagnose lymphatic diseases; without feature selection the accuracy rate was 81.2%. In line with Azar et al.’s study, Nguyen et al. [27] proposed using the Bayesian Probability to select and rank the feature weights for predictive breast cancer diagnosis. Thereafter, they utilized the random forest algorithm to prepare the model and reached an accuracy rate of 99.8%. On the basis of the above studies, it was concluded that effective feature selection in this study would improve model generalizability and accuracy.

## 3. Research Method

### 3.1. Study Design

The research process of this study was composed of seven parts, as shown in Figure 1. First, (1) database specifications were developed based on the diagnostic indicators of the assistive device evaluation report and the data were added into the database. The process then progressed to (2) data preprocessing. After data preprocessing, (3) essential features and decision-making paths were found through a decision tree. Then, (4) an evaluation of the accuracy and recall of performance was conducted. Next, (5) the data were set after feature selection was modeled into them through the CART, random forest, and Naïve Bayes algorithm. Finally, (7) the performance of the three models was evaluated and the best algorithm was selected as the predictive classification model for the LINE BOT.

### 3.2. Data Collection

The 2017–2019 evaluation year reports for wheelchairs, provided by an assistive device center in Taiwan was collected as the dataset for the study. Due to the need for personal data protection, the assistive device center first hid identities associated with the medical records in advance so that the cases could not be traced and then stored the assessment reports in the database for further manipulation. The assessment reports for “wheelchair” assistive devices are shown in Appendix A, including variables associated with feature, data type, and field number.

### 3.3. Machine Learning Algorithms

The emergence of machine learning, automating the computer system and allowing it to learn and improve from diverse experiences without being programmed, has shown great promise for the next generation of more capable artificial intelligence solutions and marks another stage in the evolution of prediction models [28]. Given the examples or instructions that humans offer, machine learning algorithm can teach computers how to learn by giving them access to data and allowing them to utilize them for learning [28].

From the viewpoint of dimensionality reduction, feature selection, in terms of better learning performance, higher learning accuracy, lower computational cost, and better model interpretability, was adopted to maximize relevance and minimize redundancy from the model. Modified feature selection method from Jović et al. [29], the machine learning pipeline is composed of three steps, which are processes toward selecting the most consistent, relevant, non-redundant features, as shown in Figure 2.

First, due to the outcomes being clearly defined and labeled, unsupervised feature selection methods are not suitable for the study. Secondly, based on the interaction with the learning model, supervised feature selection methods, are classified into three types, including filter, wrapper, and embedded methods [29]. Finally, generally, in the filter method, subsets of features based on their relationships with the target variable using statistical measures were selected; the wrapper method searched for well-performing subsets of features by model performance scores; the feature selection algorithm was automatically integrated as part of the learning algorithm using decision trees with embedded techniques.

Given the datasets of the assessment reports for “wheelchair” assistive devices in terms of missing values on “Without handicap handbook”, “Pelvis”, “Spine”, “Knee”, “Ankle”, “Cognitive ability”, “Judgment ability”, “Visual perception ability”, “Emotion control”, “Hazard level”, “The pelvis status when in a sitting position”, and “The body will fall backward when not supported”, this study adopted CART to construct the decision tree. For the purposes of simplification and purity, the questions in the assessment report were designed as “yes” or “no”. In accordance with the above design, this study adopted the binary split in CART as the classifier to find the best dividing feature among all the features. In CART, the GINI Index calculates the amount of probability of a specific feature that is classified incorrectly when selected randomly, as shown in Formula (1) [30,31] below:
(1)$$GINI\left(t\right)=1-\overset{n}{\underset{j=1}{\sum }}{\left[p(j|t)\right]}^{2}$$

*n*: Total number of categories

p(j|t): Frequency of occurrence of category *j* at node *t.*

To judge the importance of features and the interaction between different features, Breiman, in 2001, proposed the random forest ensemble learning algorithm, in which the bootstrap aggregating method is used to construct a classifier by sampling and returns the original training data. The random forest algorithm is shown below [32]:Divide the training data into N sub-training sets through bagging to collect N decision trees.Carry out the growth of the decision tree. Each node is divided by randomly selected features, and each tree grows to be complete without pruning until all features are run out.Integrate the outputs of all classifiers and then perform a majority vote or average the outputs.

Naïve Bayes model calculates the conditional probability of each sample occurring under different feature combinations according to the Bayes Theorem and selects the one with the highest conditional probability as the predictive category [14]. Basically, the Bayesian theorem is a probability model that mainly describes the probability of calculating a specific event under certain known conditions. Its formula is as follow (2) [33]:
(2)$$P\left(X|Y=y\right)=\overset{n}{\underset{i=1}{\prod }}\;P\left(X_{i}|Y_{j}\right)$$

*X*: Feature attribute set, which contains *n* attributes.

*Y*: A collection of all categories, including *y* categories.

### 3.4. Performance Evaluation

In general, accuracy and recall are adopted as model performance evaluation for machine learning [34,35]. Accuracy refers to the ratio of correctly classified samples to the total number of samples. In addition, the recall rate is how many samples are correctly predicted in all positive class samples. Both formulae are shown in (3) and (4) [34,35] below:
(3)$$Accuracy=\frac{TP+TN}{TP+TN+FN+FN}$$
(4)$$Recall=\frac{TP}{TP+FN}$$

True positive (*TP*): The actual situation is a “positive category” and the predicted result is also a “positive category”.False negative (*FN*): The actual situation is a “positive category,” but the predicted result is a “negative category”.True negative (*TN*): The actual situation is a “negative category” and the prediction result is also a “negative category”.False positive (*TP*): The actual situation is a “negative category,” but the predicted result is a “positive category”.

For generalization purposes, K-fold cross-validation is an indicator commonly used in machine learning [36]. The concept involves cutting the dataset into K parts, taking the K-1 part as the training set and the remaining 1 part as the verification set until K iterations are completed and calculated to produce a performance index, such as precision and recall [37]. Following the suggestions of Rodriguez et al. [36], this study utilized a 10-fold cross-validation with a lower evaluation error rate for model evaluation.

## 4. Data Analysis

### 4.1. Data Preparation and Preprocessing

A total of 365 assessment reports on “wheelchair” assistive devices were collected in this study. These reports fall into two broad categories: (1) “transit wheelchairs” (wheelchair type B + additional function A, referred to as B/A+), (2) “reclining and tilting wheelchairs” (wheelchair B + additional function A + additional function B, referred to as B/A+B+, wheelchair B + additional function A + additional function C, referred to as B/A+C+). For the purpose of simplification, B/A+B+ and B/A+C+ were merged into the same group. Thereafter, the numbers of “transit wheelchair” and “reclining and tilting wheelchairs” were 121 and 244, respectively. To avoid the data imbalance, in terms of the proportion of “transit wheelchairs” and “reclining and tilting wheelchairs”, which may cause model predictions to be biased toward types with a large number of samples rather than correctly identifying types with a small number of samples [38], this study adopted the suggestion of López et al. [39] and adjusted the “transit wheelchair” sample weight to 1.4. Moreover, the records followed the minimum samples requirement presented by Beleites et al. [40] for each category of the predictive classification model.

Data preprocessing is an essential part of machine learning [12,41]. Its primary purpose is not only to convert datasets into effective and complete formats, but also to help machine learning algorithms analyze, avoid operation errors, and improve the model’s overall accuracy. The following presents the preprocessing method of the data in this study:

#### 4.1.1. Missing Value Handling

(1)Replace the blank value of “Without Handicap Handbook” because there is no new handicap classification for them.(2)Replace the blank values of “Hazard level,” “The pelvis status when in a sitting position”, and “The body will fall backward when not supported” with “normal”, because the above three characteristics belong to the second layer of problems. In this way, when the first-level question is “normal”, the second-level question does not need to be answered.(3)“Pelvis”, “Spine”, “Knee”, “Ankle”, “Cognitive ability”, “Judgment ability”, “Visual perception ability”, and “Emotion control”, associated with blank value were replaced with “cannot be measured”. According to evaluation experts, the case may be bedridden due to illness, unconsciousness, and other factors. Consequently, the evaluation cannot be performed.

#### 4.1.2. Feature Deletion

Following the purpose of this study, the assistive device evaluation model was combined with a LINE BOT, an embedded AI chatbot, so that cases or their family members could engage in self-evaluation via the questions provided by the chatbot system. Some features of body size, including elbow height, hip to knee length, knee height, knee angle, popliteal angle, shoulder width, chest width, hip width, head height, shoulder height, and high subscapular angle should be precisely measured by assistive device evaluators. The feature of body size measurement (field no. 2-5-B-1 to 2-5-B-11, 2-5-b1 to 2-5-b2) was deleted mainly because of the difficulties for the cases to measure themselves; therefore, they were not included in model training. The remaining features are all important, so widely used feature selection techniques, such as random forest methods based on recursive feature elimination (RFE) and correlation analysis, are not applied in this study.

The new system of physical and mental disability identification was launched in Taiwan on 11 July 2012. The main differences between the new and old system focus on the identification of items and personnel. The classification of physical and mental disorders has been changed from 16 categories classified by disease to 8 categories classified by body structure and function, in terms of activity participation and environmental factor. Meanwhile, instead of the doctor’s appraisal, a certificate of physical and mental disability would be issued by medical and professional teams after medical appraisal and needs assessment in the new system. Due to the lack of the disability classification bridge between the Disability Handbook Category of the old version and the new version, information associated with the feature in the new version could not be used and was deleted for further analysis.

#### 4.1.3. Data Transformation

This study dataset contains several multi-category feature attributes, and binary coding is required for this feature type in advance. In this sense, for example, “Usage purpose and activity requirements” is a selectable feature, including daily life, medical care, schooling, employment, and leisure activities (field no. 2-1) and belongs to a multi-category attribute. After binary encoding as shown in Table 1, each attribute value is uniquely assigned to facilitate model training.

### 4.2. Decision Tree Analysis

For the purpose of cross-validation, “transit wheelchair” and “reclining and tilting wheelchairs” were taken as the classification targets in this study, the dataset was divided into an 80.0% training subset (292 pieces of data) and a 20.0% (73 pieces of data) testing subset, and the testing set was used for performance evaluation. By gleaning decision rules from the above dataset, the results revealed that there were four layers and five nodes, including head condition, age, pelvic condition, cognitive ability, judgment ability, in the decision tree. Meanwhile, the recall rate for “transit wheelchair” was 76.9%, the recall rate of “reclining and tilting wheelchairs” was 87.2%, and the over overall model accuracy rate was 83.6%.

For the purposes of making it simple to understand, the hierarchical structure of each path is elaborated on Figure 3:

A.When the case “head movement control status” is abnormal, then, go to the left subtree:
(1)A total of 99 out of 112 samples (88.4%) belong to “reclining and tilting wheelchairs”.(2)When the “cognitive ability” of the case was normal, 66 of the 71 samples (92.9%) belonged to “reclining and tilting wheelchairs”.(3)When the “cognitive ability” of the case was abnormal, 33 of the 41 samples (80.5%) belonged to “reclining and tilting wheelchairs”.B.When the case’s “head movement control status” is normal then, go to the right subtree:
(1)A total of 98 of the 180 samples (54.4%) belonged to the “transit wheelchair”.(2)When the “age” of the case is less than 71 years old (the node is represented as 70.5), the GINI coefficient is 0.33, and it belongs to the “transit wheelchair” category.C.In cases where the “head movement control status” is normal and the “age” is higher than 71 years old:
(1)When the “pelvic condition” of the case was abnormal, 35 of the 44 samples (79.5%) belonged to “reclining and tilting wheelchairs”.(2)When the “pelvic condition” of the case is normal, the GINI coefficient is 0.5, which belongs to “reclining and tilting wheelchairs”.D.When the case’s “head movement control status” is normal, “age” is higher than 71 years old, and “pelvic condition” is normal:
(1)When the “judgment ability” of the case is abnormal, the GINI coefficient is 0.5, and it is mainly a “transit wheelchair”.(2)When the “judgment ability” of the case is normal, the GINI coefficient is 0.498, which mainly belongs to “reclining and tilting wheelchairs”.

In terms of predictability, by analyzing the results of the decision tree, it is possible to learn the main symptoms associated with the wheelchair types. These are described below:
A.Transit wheelchair:

The “head movement control” is normal, and the “age” is less than 71 years old. This indicates that the cases with a lower “age” have a lower probability of relative severity of the symptoms and are more able to maintain a normal sitting position without needing to lie on the back or tilting for assistance.

B.Reclining and tilting wheelchairs:
(1)“Head movement control” is abnormal. One possible explanation related to this is that when the head cannot maintain a fixed position naturally, the head would be tilted, which might cause physical discomfort or deformation in the long term. Therefore, the function of lying on the back or tilting is required to adjust the sitting posture.(2)“Head movement control” is normal, “age” is higher than 71 years old, and “pelvic condition” is abnormal. The reason, in terms of the above scenarios, is that if the pelvis is tilted forward, backward, or offset, it would lead to a poor sitting posture and the case would need “reclining and tilting wheelchairs” for assistance.

C.Transit wheelchair or reclining and tilting wheelchairs:

“Head movement control” is normal, “age” is higher than 71 years old, “pelvic condition” is normal, and “judgment ability” is normal. It is difficult to suggest a wheelchair type using the case’s “judgment ability” as a node. Surely, assistive device evaluators should go to the scene to make closer observations and give more appropriate recommendations.

### 4.3. Model Comparison

For the purpose of quantitative performance evaluation in terms of accuracy and recall with a 10-fold cross-validation, “without feature selection” and “with feature selection” derived from decision tree were further compared on CART, Naïve Bayes, and the random forest algorithm for model building.

Table 2 revealed that the recall rate of “Transit wheelchair” without feature selection was low with 56%, 45.4%, and 68.7%, respectively. The results also reflected that the accuracy of the random forest without feature selection reached 73.0%, which was the highest among the three models. In a similar vein, this study echoed results from Azar et al.’s and Nguyen et al.’s study that effective feature selection would improve model generalizability and accuracy. Feature selection clearly not only improved model accuracy and “transit wheelchair” recall, but also easily provided answers to cases or their family members when these features were transferred into questions embedded into the AI chatbot (LINE BOT).

Broadly speaking, SVM is suitable for numerical data, yet BN requires the help of experts to build an effective network [15]. The datasets of the assessment reports for “wheelchair” assistive devices (shown in Appendix A) have category types and multi-class characteristics typified by the lack of nonlinear, binary characteristics or professional knowledge from experts. Thus, in line with the data type associated with the categorical data of this study, and given the highest model accuracy of 72.0%, and a recall rate for both wheelchair types of more than 70.0%, Naïve Bayes model was further adopted to handle classification prediction problems during LINE BOT operation.

### 4.4. Application Interface Design for Users

With ease in practical implementation in mind, the five key features, including head condition, age, pelvic condition, cognitive ability, and judgment, derived from the decision tree algorithm were further discussed with assistive device evaluation experts. According to experts’ suggestions, “cognitive ability” and “judgment ability” are often used in practice as the standard of cognitive ability, so these two characteristics were regarded as the same and combined into one. Thereafter, four questions were designed for users to answer (as shown in Table 3). Using smart self-service to provide instant solutions, a LINE BOT was designed to allow “cases or their family members” to self-assess and inquire about resources related to assistive devices.

Cases can answer the question by clicking the button below the picture, as shown in Figure 4 and Figure 5. After the user answers the four questions, the LINE BOT would analyze it through the Naïve Bayes prediction model, give suggestions on the type of wheelchair, and display the picture, as shown in Figure 6. To bridge the gap in understanding the case’s status, the response and type suggestion would be simultaneously sent to the assistive device center as a reference to help evaluators prepare appropriate assistive device types.

## 5. Conclusions and Research Suggestions

### 5.1. Conclusions

In Taiwan, current assistive device assessment mainly uses telephone consultation to learn the status of the case. It is typically difficult to determine the individual’s physical condition and the corresponding assistive device type that should be employed. Through the LINE BOT designed by this study, the case or the case’s family can easily use the LINE BOT for self-assessment during the telephone consultation. The LINE BOT generates insights for not only enabling cases to initially understand their own assistive device “type” needs, but also provides evaluators with information and suggestions on the case’s physical condition, allowing them to prepare appropriate assistive devices for the case to try and in turn reducing the time cost associated with field assessments.

### 5.2. Research Suggestions

In this study, the machine learning algorithm was used to capture five key evaluation features and the model accuracy rate was up to 72.0%, but there is still much room for improvement. The suggestions of this study as regards data and application are provided below.

#### 5.2.1. Data

(1)This study only analyzed 365 diagnostic records from a specific assistive device center. Although the minimum number of samples for model classification is reached, future studies should add more samples so as to enhance the model’s classification accuracy and more completely assess generalizability of findings.(2)In a similar sense, the study deleted some features of body size measurement. Future studies should not only consider users’ convenience but also combine these features with a LINE BOT to allow the users or the users’ family to report their situation.(3)From the viewpoint of feature selection, the researchers should keep in mind to the correspondence table between the new system (8 categories) and the old system (16 categories) of disability categories and codes has been amended.(4)This study is only for the prediction and classification of “transit wheelchairs” and “reclining and tilting wheelchairs”. With numerous calls for wheelchairs in an “aging society”, further research can include other wheelchair types to enrich the precision of results.

#### 5.2.2. Application

From a practical perspective, the increasing popularity and the timing of LINE BOT use are two core issues. The time and cost of advice and information on assistive devices could obviously be reduced if the case or family members of the case are known in the early stage of the assessment process. Therefore, this study recommends the following two situations for the use of the LINE BOT:
(1)At its most basic, when the patient is ready to be discharged from the hospital, the hospital can physically provide the patient to use the assessment assistive device, LINE BOT, on his/her mobile phone. Essentially, this enables the patient to learn the type of assistive device they need during the referral of the assistive device center at the hospital and notify the assistive device center to prepare for follow-up assessment services.(2)When calling the assistive device center for inquiry, the case can be reminded to join the LINE BOT with his/her mobile phone for preliminary self-assessment during telephone consultation. Moreover, the QR Code of the LINE BOT can be added to the webpage of the assistive devices center, and provide case scanning in order to link LINE BOT in a cyberspace environment.

## Figures and Tables

**Figure 1 healthcare-10-02238-f001:**
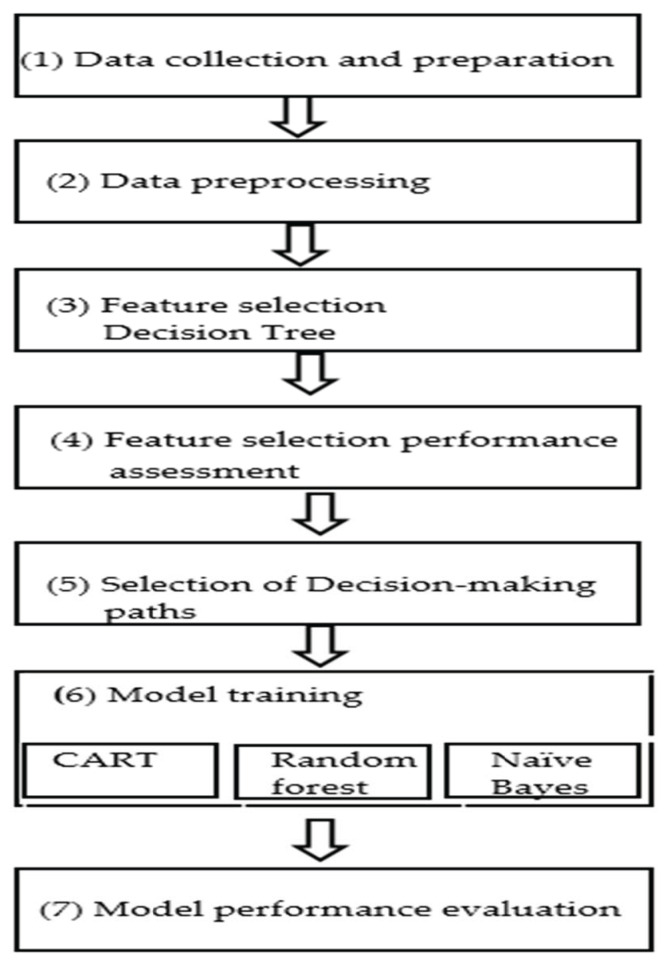
Research process of the study.

**Figure 2 healthcare-10-02238-f002:**
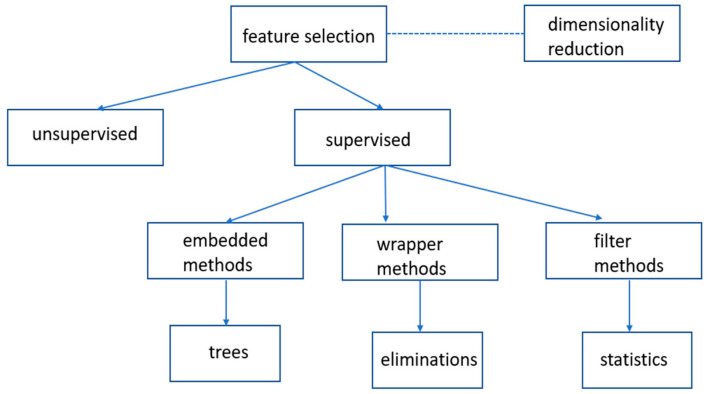
Feature selection techniques.

**Figure 3 healthcare-10-02238-f003:**
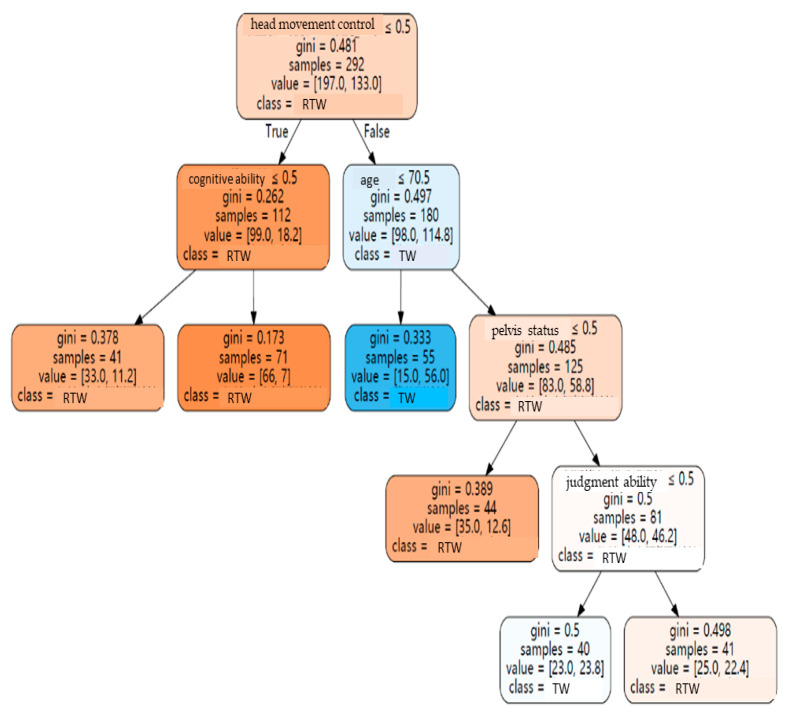
Decision tree path diagram. Note: 1. Samples represent the number of cases of this node; value represents the number of “reclining and tilting wheelchair” samples, and the number of “transit wheelchair” samples*1.4 (sample weight); class represents the classification result of this node. 2. RTW = reclining and tilting wheelchair, TW = transit wheelchair.

**Figure 4 healthcare-10-02238-f004:**
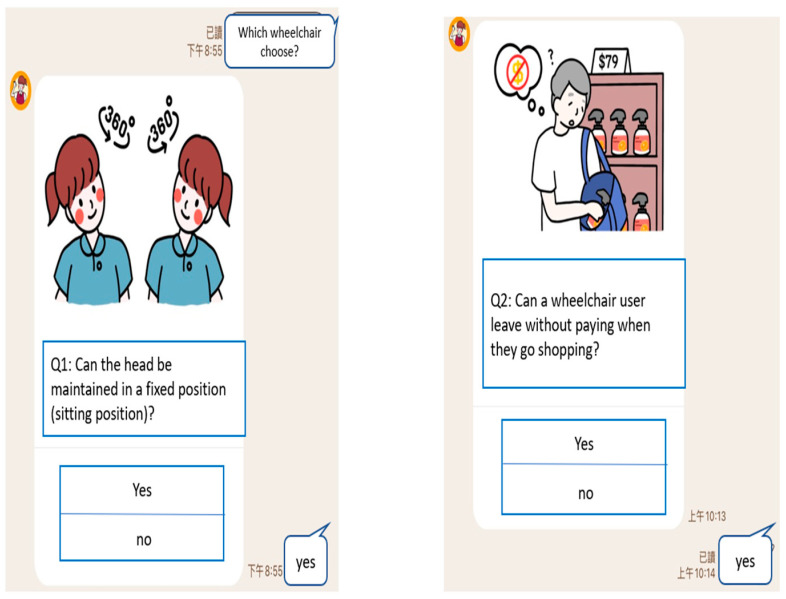
Assistive devices helper (LINE BOT)—question 1 (**left**)/question 2 (**right**).

**Figure 5 healthcare-10-02238-f005:**
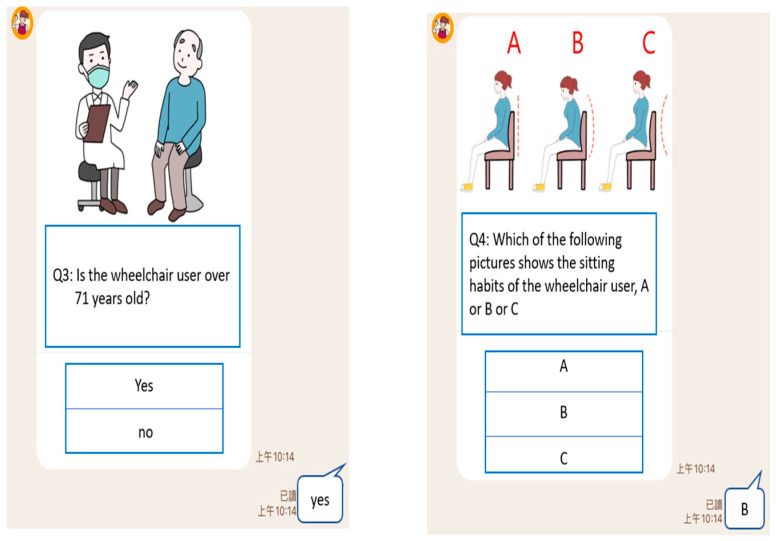
Assistive devices helper (LINE BOT)—Question 3 (**left**)/Question 4 (**right**).

**Figure 6 healthcare-10-02238-f006:**
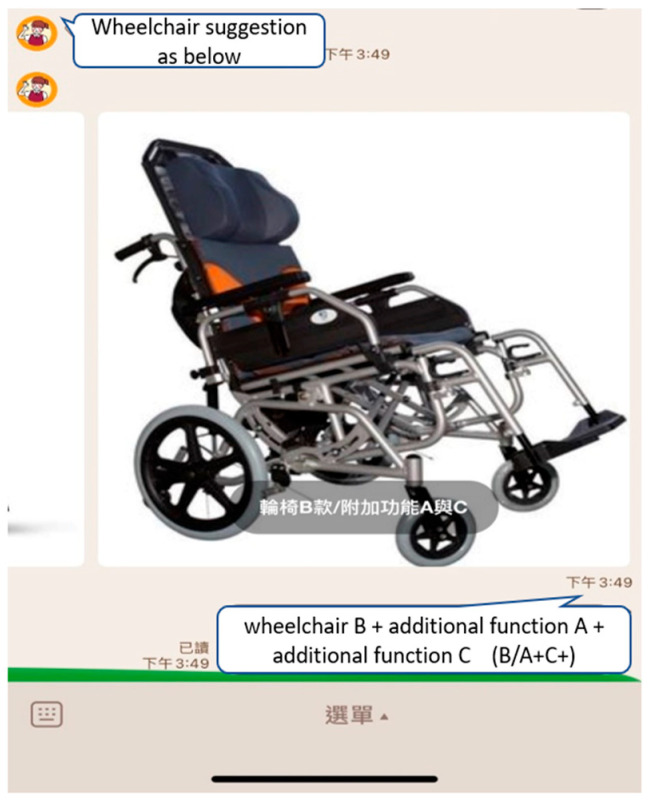
Assistive devices helper (LINE BOT)—Wheelchair type suggestion.

**Table 1 healthcare-10-02238-t001:** Examples of use purposes and activity requirements.

Before Encoding	After Encoding
Daily life	1
Medical	10
School	11
Work	100
Leisure activities	101

**Table 2 healthcare-10-02238-t002:** Performance comparison of three models.

	Feature Selection	Accuracy	“Transit Wheelchair” Recall Rate	“Reclining and Tilting Wheelchairs” Recall Rate
CART	yes	70.70%	63.70%	74.20%
	no	68.50%	56%	74.60%
Naïve Bayes	yes	72.00%	71.20%	72.60%
	no	68.70%	45.40%	80.10%
Random forest	yes	70.90%	72.00%	70.50%
	no	73.20%	68.70%	75.40%

**Table 3 healthcare-10-02238-t003:** LINE BOT assessment questions.

Decision Tree Node (Feature)	Question	Reply Option
Head movement control	Q1: Can the head be maintained in a fixed position (sitting position)?	Yes: normal head movement control No: abnormal head movement control
Cognitive ability	Q2: Can a wheelchair user leave without paying when they go shopping? (If the wheelchair user is confused or unable to answer, please write “Yes”)	Yes: poor cognitive and judgment skills No: normal cognitive and judgment skills
Judgment ability		
Age of 71	Q3: Is the wheelchair user over 71 years old (inclusive)?	Yes: age over 71 No: younger than 71 years old
Pelvis status	Q4: Which of the following pictures shows the sitting habits of the wheelchair user, A or B or C?	A: pelvis normal B: posterior pelvic tilt, abnormal C: anterior pelvic tilt, abnormal

## Data Availability

The datasets used and/or analyzed during the current study are available from the corresponding author on reasonable request.

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
