# Peer review of "Using Machine Learning to Explore the Crucial Factors of Assistive Technology Assessments: Cases of Wheelchairs"

_healthcare, 2022, doi:10.3390/healthcare10112238_

Round 1
Reviewer 1 Report
The paper is well written and establishes an evaluation model. Furthermore, users can self-assess through the model’s initial advice on wheelchair type selection. This research enables evaluators to determine a case’s situation in advance, reduce the time required for fixed-point or home assessments, and carry the appropriate size of assistive devices.
However, there are some issues that must be solved before it is considered for publication. If the following issues are well-addressed, this reviewer believes that the essential contribution of this paper is significant for assistive technology assessments, especially in wheelchairs. These problems are given below.
Structure:
1. The third section of 4.3 compares SVM and NB and concludes that NB is more suitable for classification prediction problems. This paragraph is proposed to be combined with the first two paragraphs of 2.3.
2. It is suggested that the experimental arguments of others in the last paragraph of 2.3 should be combined with the table data in 4.3. model comparison. Finally, it is concluded that effective feature selection in this study would improve model generalizability and accuracy.
3. Paragraph 4 and 5 of 2.3 compares CART with C4.5 and ultimately selects CART with higher accuracy. The 3.3machine learning algorithm section mentions the ID3, C4.5, and CART algorithms but does not analyze them and compare them to choose CART directly. Therefore it is recommended that the two parts be combined.
4. 4.2 Decision tree analysis section judges wheelchair types based on the values of the five essential characteristics, while 5.1. the conclusion section describes the main symptoms associated with wheelchair types. It may be better if the reader understands the attributes of wheelchair type before making judgments about wheelchair type, so I would suggest that 5.1. the conclusion section is placed before the 4.2 decision tree analysis.
Results:
5. 4.2.Decision tree analysis has some incorrect language descript-
tions, corrections as follows:
â… .LINE.366:The word“poor” should be changed to“normal.”
â…¡.LINE.368:The word“normal” should be changed to“abnormal.”
â…¢.LINE.372:The phrase“‘reclining and tilting’ wheelchair” should be changed to“‘transit’ wheelchair.”
â…£.LINE.385:The word“normal” should be changed to“abnormal.”
â…¤.LINE.387:The phrase“confused or slow” should be changed to“normal.”
Suggestions:
6. The first part of 5.2.3. application may be better if the hospital provides the patient's five key characteristics data directly to the LINE BOT for evaluation, resulting in the recommended wheelchair type. At the same time, the recommended wheelchair type will be sent to the Assistive Device Center staff to prepare appropriate assistive devices.
Author Response
- Thanks for the reviewer’s comments and suggestions.
- Authors have tried to response and provide line reference (please check the attached file):

Reviewer 2 Report
Thank you for the opportunity to review a novel study and interesting manuscript! The text is well-structured, clear and easy to follow, but I would like to make the following suggestions to improve the manuscript:
Consider using "assistive products" instead of "assistive products" throughout to align the text to ISO 9999 and World Health Organization terms.
Use the same number of decimals in all %-numbers from section 2 and onwards. (For example, use one decimal in all numbers - also where it is 0.)
In the Abstract, the purpose of the study is not clear. Please align it explicitly with the purpose in section 1.
Row 10: Replace "disabilities" with "limitations"
41: Replace "demand of" with "demand for"
42-46: Provide substantiating references. (Also, avoid using "Without a doubt")
48-51: Indicate between what years the increase occurred
88: Replace "important part" with "path"
104-105: Avoid "elderly" as it reflects ageism. Use "older people" throughout instead
106: Delete "No doubt"
106-107: Avoid "disabled" and use "people with disabilities" throughout instead
106-108: Provide substantiating reference
122: Describe what you mean by "assessment assistive devices"
190: Add "and" after "performance" and delete "has"
197: Add "with" after "line"
415: Explain how the questions were designed and what kind of testing of them was performed
Appendix A: Harmonize the use of capital letters across all variables and data types (for example, "The old system" vs. "The new System"; "isomorphism" vs. "Isomorphism")
References: Ensure that the correct format is used for each and every reference. For example, the journal title of 9 is abbreviated but not 8; Semicolon between authors is used in 9 but not 8.
Author Response
- Thanks for the reviewer’s comments and suggestions.
- Authors have tried to response and provide line reference (please check the attached file)
